# Land Subsidence Control Zone and Policy for the Environmental Protection of Shanghai

**DOI:** 10.3390/ijerph16152729

**Published:** 2019-07-31

**Authors:** Xi-Cun He, Tian-Liang Yang, Shui-Long Shen, Ye-Shuang Xu, Arul Arulrajah

**Affiliations:** 1State Key Laboratory of Ocean Engineering, School of Naval Architecture, Ocean, and Civil Engineering, Shanghai Jiao Tong University, Shanghai 200240, China; 2Key Laboratory of Land Subsidence Monitoring and Prevention, Ministry of Land and Resources & Shanghai Engineering Research Center of Land subsidence, Shanghai 201204, China; 3Department of Civil and Environmental Engineering, College of Engineering, Shantou University, Shantou 515063, China; 4Department of Civil and Construction Engineering, Swinburne University of Technology, Melbourne 3122, Australia

**Keywords:** sustainable environment protection, land subsidence, management regulation

## Abstract

Land subsidence was once a major geo-hazard in the city of Shanghai, China. From 1921 to 1965, the maximum cumulative land subsidence in the urban areas of China reached 2.6 m. This large subsidence has resulted in high economic losses for Shanghai. The Regulation of Prevention and Control of Land Subsidence of Shanghai Municipality was published in 2013 (simply cited as the 2013-regulation in the following context). The characteristics of the 2013-regulation included the combination of the subsidence monitoring network and the groundwater detection network due to both the effects of groundwater withdrawal and construction. In addition, the setting up of a supervision system was also incorporated in the 2013-regulation. To control the land subsidence, Shanghai demarcated three land subsidence control zones, where special measures have been implemented. From a strategic environmental assessment (SEA) point of view, the 2013-regulation attains a high total score, indicating that the control of groundwater withdrawal and recharge is effective. The observed land subsidence over the past six years also confirms the effectiveness of the 2013-regulation with the most consideration of SEA for sustainable environment protection in Shanghai. However, more effort should be made in the implementation of SEA in land subsidence control in the future.

## 1. Introduction

Land subsidence has been observed in many large metropolitan cities such as Shanghai [1,2,3], Tianjin [4,5,6], Xi’an [7], Houston [8], Bangkok [9], and Mexico [10]. Land subsidence results in loss of altitude, invasion of seawater, and urban waterlogging [11,12,13,14,15,16,17,18]. Non-uniform land subsidence results in damages to structures such as underground tunnels and elevated roads [19,20]. With the improvement in environmental protection, cities are increasingly monitoring and controlling land subsidence. A variety of control policies for land subsidence has been established in the cities suffering land subsidence. The land subsidence in Bangkok once reached more than 1 m over the past 35 years [9]. In trying to mitigate groundwater withdrawal, the Bangkok government decided to raise the price of groundwater at the turn of the century. Mexico City is another well-known city that suffers from serious land subsidence. The regional Interferometric Synthetic Aperture Radar with Small Baseline Interferometry Time-series Techniques (InSAR-SB time-series) survey was conducted by researchers to monitor the land subsidence in central Mexico [10]. The monitoring system can provide guidance for groundwater recharge/withdrawal balance. In Houston, a large number of GPS stations were installed to study the ground deformation induced by land subsidence [8]. A monitoring network based on GPS was installed to collect continuous data [21].

The monitoring of land subsidence in Shanghai can be first traced back to the beginning of the 20th century. From 1921 to 1965, the maximum cumulative land subsidence in urbanized districts reached 2.6 m (nearly 59 mm per year). Since 1966, the Shanghai government has realized that land subsidence can cause geo-hazards and significant economic losses. Groundwater withdrawal in urban areas was forbidden. Until 1985, the land subsidence in urban areas was well controlled (only 1 mm per year). At the same time, the land subsidence in suburban regions was evident since the groundwater pumping. However, with the rapid development of China’s economy and society, land subsidence in both urban and suburban regions began to deteriorate after 1986 [22]. The implementation of multiple measurements since 2000 has reduced the annual land subsidence to below critical values. In 2006, the Regulation of Prevention and Control of Land Subsidence of Shanghai Municipality (labeled as the 2006-regulation) was implemented. But the 2006-regulaton did not follow the principles of strategic environmental assessment (SEA) for sustainable urban development [23]. Most of the principles of SEA were just partially considered in the 2006-regulation. Considering the deficiency of 2006-regulation, the Regulation of Prevention and Control of Land Subsidence of Shanghai Municipality in 2013 (labeled as the 2013-regulation) was issued [24,25]. To ensure the sustainable development of Shanghai, the implementation effect of the 2013-regulation should be evaluated after 6 years of application.

The intent of this paper is to introduce the characteristics and measurements of the 2013-regulation and evaluate the effect of the implementation of the 2013-regulation on the control of land subsidence in Shanghai. First, the situation and reasons of land subsidence in Shanghai is introduced. Then the history of subsidence management in Shanghai is presented. Based on the characteristics of the 2013-regulation, the specific measures of land subsidence in Shanghai is also introduced. Lastly, the implementation effect of the 2013-regulation is discussed, based on the SEA level.

## 2. Situation and Reasons of Land Subsidence in Shanghai

Figure 1 shows the overall distribution of land subsidence in Shanghai from 1921 to 2016 [18]. The largest land subsidence in urban areas was up to 2.6 m (Figure 2). Until now, the annual land subsidence in Shanghai is controlled within 6 mm [26]. Groundwater withdrawal is the main reason that results in land subsidence in Shanghai. Figure 2 is a profile that shows the geology and hydrogeology information of Shanghai. The land in Shanghai is composed of soft deltaic sediments with several isolated outcrops of bedrock [23]. Most of the bedrock is buried under Quaternary sediments [27]. The aquifers in Shanghai consist of a phreatic aquifer group (Aq0) and five confined aquifers (AqI to AqV), of which AqII to AqV are the groundwater withdrawal layers in Shanghai [28,29]. Aq0 is not represented in Figure 2.

Shanghai is a coastal city that has 213 km of coastlines. With the rapid changes in sea-level and coastal land subsidence, the risk of coastal hazards such as marine flooding, storm surges, and tsunamis, will increase annually [30,31,32]. Figure 3 shows the global average sea level from 1990 to 2009. As the change in global climate continues, the sea level is rising fast and is expected to continue rising for centuries [33,34]. In the Shanghai coastal area, the mean annual eustatic sea level of Yangtze River estuary reached 2.5 mm from 1997 to 2010 [35,36]. From 1997, the sea level along Shanghai will likely rise 86.6 mm, 185.5 mm, and 433.1 mm by 2030, 2050, and 2100, respectively [36]. Meanwhile, ongoing coastal land subsidence will amplify sea-level rise [36]. With the combination of coastal land subsidence and sea-level rise, the risk of coastal hazards will be higher. Since land subsidence will cause many problems with urban infrastructure and coastal hazards, control measures should be adopted to prevent damage induced by land subsidence.

## 3. History of Subsidence Management in Shanghai

Figure 4 shows the publishing time of main management policies for land subsidence in Shanghai over the time frame of 1920 till present day. The land subsidence in Shanghai was first recorded in 1921. The Shanghai government first implemented the control of land subsidence in 1963, when the cumulative land subsidence had reached approximately 1.69 m. In 1963, the regulation for deep wells in Shanghai was issued. In 1966, the artificial recharge was conducted in the urban area, based on the policy named “winter-recharge and summer-withdraw”. At the end of 1960s, the land subsidence in urban area was generally controlled. Following the initial success, the government further enhanced the groundwater recharge capacity, and the withdrawal of groundwater was adjusted to the deep aquifers [37]. With the urbanization of Shanghai, a large number of factories were removed from the urban area to the suburban region. Most of deep wells in the urban area were shut down during this period. In order to control increasing land subsidence in suburban areas, the municipal water supply project was conducted to reduce groundwater withdrawals.

For better control of land subsidence, Shanghai began building its own monitoring network by installing levelling points in the entire administrative region. In 1996, a regulation for the management of land subsidence monitoring network (labeled as the 1996-regulation) was approved by the municipal government. As the protection of groundwater is a regional issue in the large scale, the Shanghai municipal government persuaded Jiangsu province authorities to enhance the ban to groundwater withdrawals. In the current century, a great quantity of skyscrapers constructed in urban areas have further aggravated land subsidence. In 2003, the urban planning (labeled as the 2003-planning) proposed by the Shanghai Municipal Planning and Ground & Resources Administration, allowed the increase of a public greenbelt and space, and to decrease the plot ratio and the amount of buildings. Due to the high economic cost of land subsidence, the municipal government established a threshold of land subsidence at a value of 6 mm/year in the Twelfth Five-year Plan of subsidence management. In 2013, the latest regulation for the management of land subsidence in Shanghai was approved, and it has been successfully implemented for the past 6 years. Relying on the 2013-regulation, the land subsidence in the entire administrative region has been reduced to about 5 mm per year.

## 4. Characteristics and Specific Measurements of Regulation

The 2013-regulation was drafted based on a wide range of scientific research projects and experiences from decades of subsidence management. There are four main characteristics compared to previous regulations, which are: (i) Combining the subsidence monitoring network and groundwater detecting network, (ii) considering the effect of groundwater withdrawal and project construction, (iii) building a supervision system for the entire process, and (iv) dividing the Shanghai city into land subsidence control zones. More specific measurements related to these characteristics are presented in the following sections.

### 4.1. Combining Monitoring Networks

The 2013-regulation was proposed to improve the subsidence monitoring network based on the existing monitoring points for major municipal projects. The monitoring network uses a variety of technologies, including GPS and GIS, to automatically measure land subsidence each year. Meanwhile, the setting up of a groundwater detecting network was required for these purposes. One important measure in the 2013-regulation was to combine these two monitoring networks, which enables monitoring of the land subsidence and groundwater level in the entire administrative region, as shown in Figure 5. All the related monitoring data should be made available to the public.

### 4.2. Considering Multiple Factors

Both the dewatering of groundwater and construction of high-rise buildings are considered as important factors that contribute to land subsidence. The municipal government invests in the construction of recharge wells and makes groundwater recharge plans each year. Groundwater withdrawal has been forbidden in the region covered by the pipeline network of water supply since the application of the 2013-regulation. Based on the subsidence monitoring results, a planning area called subsidence-sensitive area is defined. Based on the current land subsidence and the development of urbanization, a prior prevention area is further designed in the planning area. The planning of urbanization in this region should adjust the development scale to prevent the further land subsidence.

### 4.3. Supervising the Entire Process

A supervision system for the 2013-regulation was developed to prevent land subsidence, monitor the process, and mitigate geo-hazards caused by land subsidence including loss of altitude, invasion of seawater, and urban waterlogging. According to monitoring results, the control objective of land subsidence should be determined each year. A series of measurements are provided to prevent the occurrence and mitigate the adverse effect of land subsidence.

In subsidence-sensitive areas, a scientific assessment is required before the excavation of foundation pits of depths exceeding 15 m. The necessary contents in the assessment include: (i) The possibility of suffering from land subsidence for the surrounding buildings, (ii) the influence degree and scale of land subsidence during and after the construction, and (iii) suggestions for the prevention of land subsidence.

During construction, monitoring networks together with newly installed detecting wells are used to conduct real-time supervision. In the maintenance period, the land subsidence near major municipal projects should be uploaded to the administrative departments. Government officers should regularly inspect the equipment in the field and provide guidance for daily operation.

If an unacceptable amount of subsidence occurs, related contingency plans will be performed immediately to mitigate all unfavorable factors. For the subsequent sanctions after the occurrence of land subsidence, the 2013-regulation defines specific legal liabilities and perfect standards in different conditions of land subsidence.

### 4.4. Dividing Land Subsidence into Control Zones

#### 4.4.1. Division of Land Subsidence Control Zone

On the basis of the 2013-regulation, Shanghai has proposed the innovative concept of land subsidence control zones, which consider the geological background, urban development planning, land subsidence susceptibility, and the development status of land subsidence. Shanghai is divided into three land subsidence control zones, including key land subsidence control zones I (containing I_1_, I_2_, I_3_), sub-key land subsidence control zones II, and general land subsidence control zones III. The main information of each land subsidence control zone, such as area, distribution areas and control objectives, are listed in Table 1. The different land subsidence control zones distributed over Shanghai are given and mapped (Figure 6).

#### 4.4.2. Main Control Measures

Different control zones have different measures to control land subsidence in Shanghai. There are five control measures in control zones I_1_, seven control measures in control zones I_2_, three control measures in control zones I_3_, three control measures in control zones II and four control measures in control zones III. The detailed control measures applied in each control zones are listed in Table 2.

## 5. Discussion

Shepherd and Ortolano [40] proposed a strategic environmental assessment (SEA) method to evaluate the environmental impact of policies, plans and programs. Xu et al. [23] used SEA to evaluate the 2006-regulation [24] for the management of land subsidence in Shanghai. The eventual results including 1996-regulation, 2003-planning, and 2006-regulation were not satisfactory. The improvement of the 2013-regulation in SEA analysis is presented in Table 3. The score of the 2013-regulation in SEA was 24, while the score of the 2006-regulation in SEA was 16 (total score is 30).

Figure 7 shows the volume of groundwater withdrawal and recharge from 2000 to 2017 in Shanghai [41]. Before the approval of the 2013-regulation, the groundwater withdrawal showed a decreasing tendency and groundwater recharge showed an increasing tendency [42,43]. After the approval of the 2013-regulation, the decrease of groundwater withdrawal and the increase of groundwater recharge were more obvious.

The 2013-regulation has been implemented for nearly 6 years since being approved in 2013. Figure 8 shows the monitoring results of land subsidence published in the Shanghai Geological Environmental Bulletin [26]. After the approval of the 2013-regulation, the land subsidence in the entire administrative region shows a decreasing tendency. All the recent monitoring results are below the limited value (6 mm/year). The effectiveness of the 2013-regulation can be confirmed by field monitoring results.

Shanghai is a city with plentiful water resources, such that land subsidence can be limited by the control of groundwater withdrawal and recharge. The 2013-regulation is formulated according to the characteristics of Shanghai city. The experience of the 2013-regulation on Shanghai can be used for reference even though it may not be directly applied to other cities. For example, it is inadvisable to limit the land subsidence by controlling groundwater withdrawal in the North China Plain due to the shortage of water resources.

## 6. Conclusions

The 2013-regulation was implemented to manage land subsidence due to sustainable urbanization. By reviewing the history of land subsidence management in Shanghai, the constitutive principle was to improve the effectiveness and sustainability of management measures. The 2013-regulation further combines the subsidence monitoring network and groundwater detecting network, considering effects of groundwater withdrawal and constructions, and building a supervision system in the entire process. Based on the characteristics of the 2013-regulation, Shanghai is divided into three land subsidence control zones, and each control zone has its special measures to control the land subsidence. According to the field data, the implementation of the 2013-regulation controlled the annual land subsidence to below the limited value. Hence, the effectiveness of SEA in the municipal’s environmental management is verified.

## Figures and Tables

**Figure 1 ijerph-16-02729-f001:**
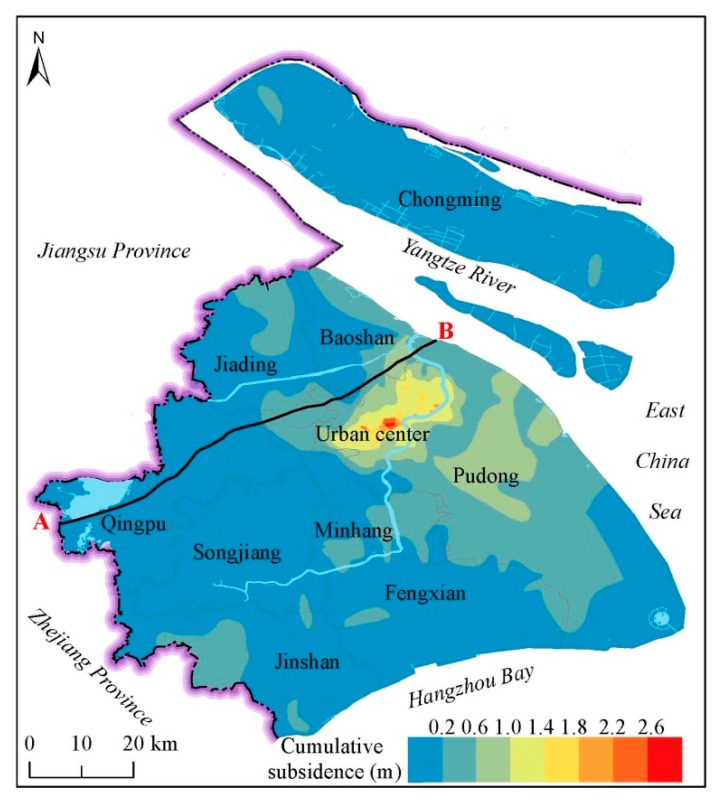
Overall land subsidence in Shanghai (from 1921 to 2016) (data from Reference [18]).

**Figure 2 ijerph-16-02729-f002:**
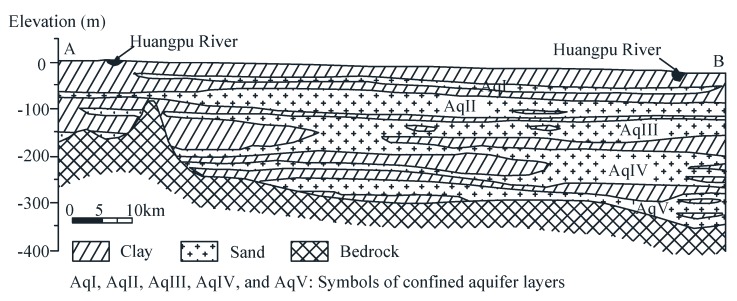
Geology and hydrogeology profile of Shanghai (cross-section A–B in Figure 1) (data from Reference [28]).

**Figure 3 ijerph-16-02729-f003:**
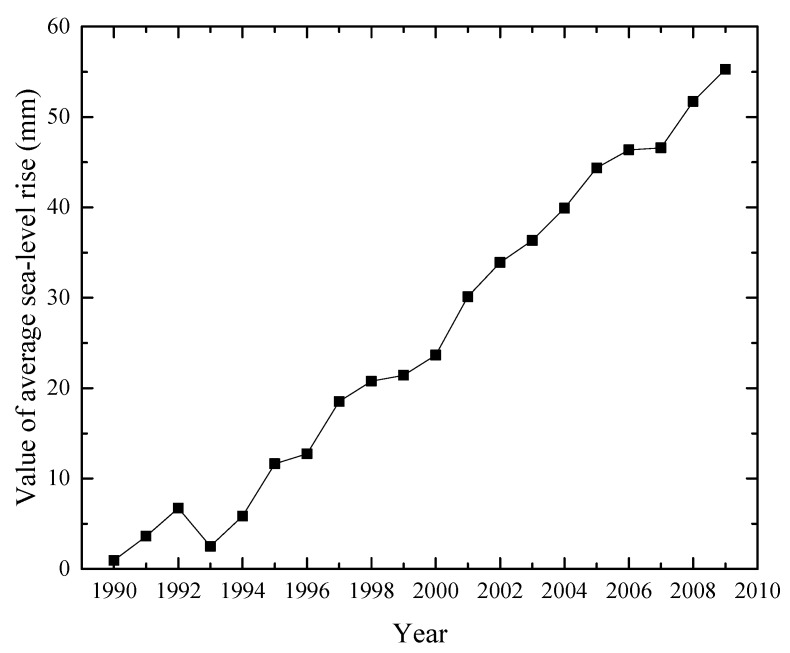
Global average sea level rise from 1990 to 2009 (Data from Reference [34]).

**Figure 4 ijerph-16-02729-f004:**
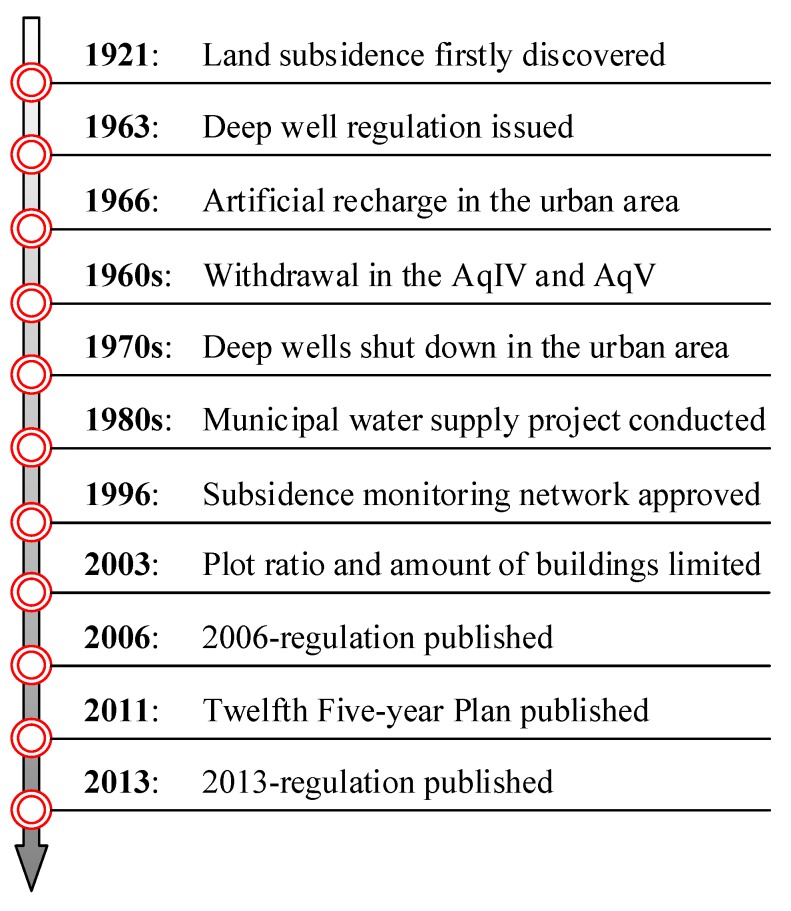
History of subsidence management in Shanghai (from 1920 to 2019).

**Figure 5 ijerph-16-02729-f005:**
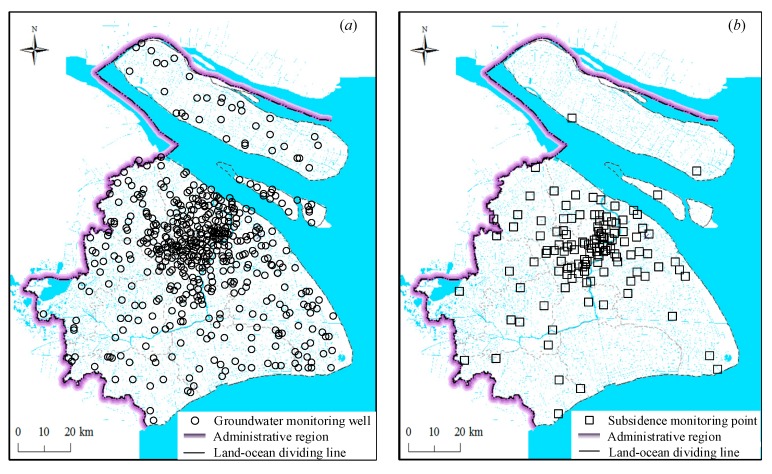
Monitoring networks in Shanghai: (**a**) Groundwater detecting network, (**b**) land subsidence monitoring network (redrawn used the data from Reference [38]).

**Figure 6 ijerph-16-02729-f006:**
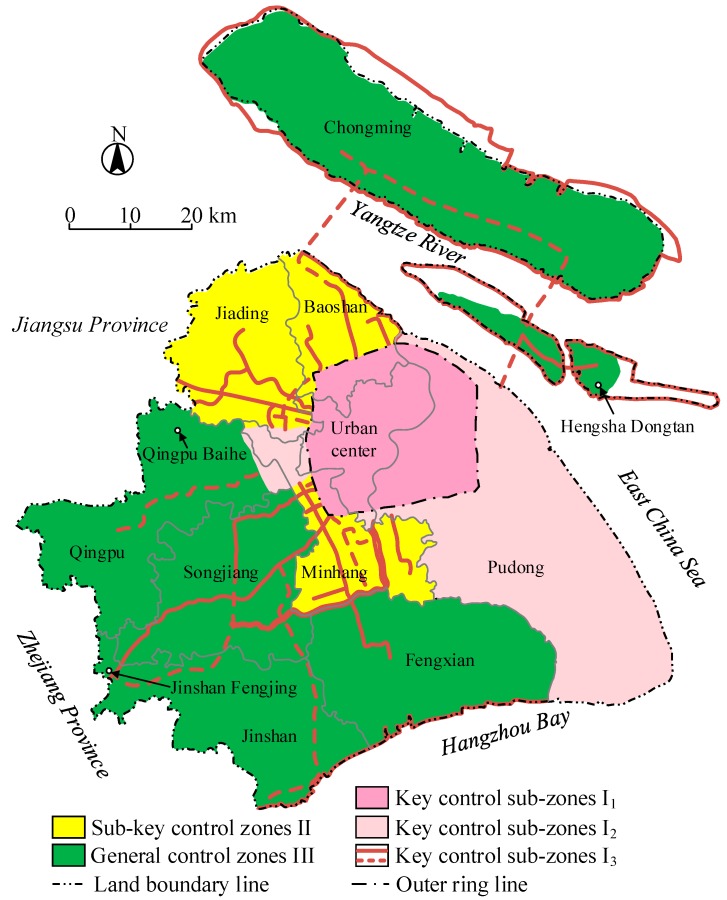
The comprehensive division map of land subsidence control zones in Shanghai (data from Reference [39]).

**Figure 7 ijerph-16-02729-f007:**
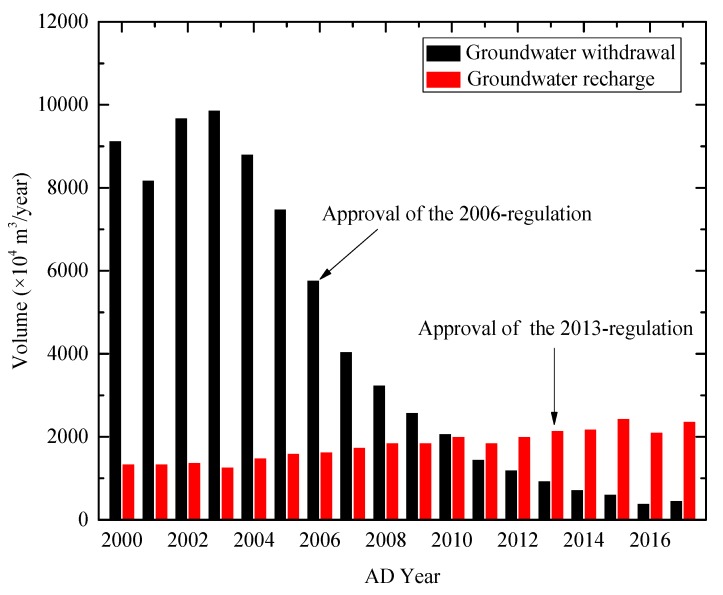
Volume of groundwater withdrawal and recharge in Shanghai from 2000 to 2017 (data from Reference [41]).

**Figure 8 ijerph-16-02729-f008:**
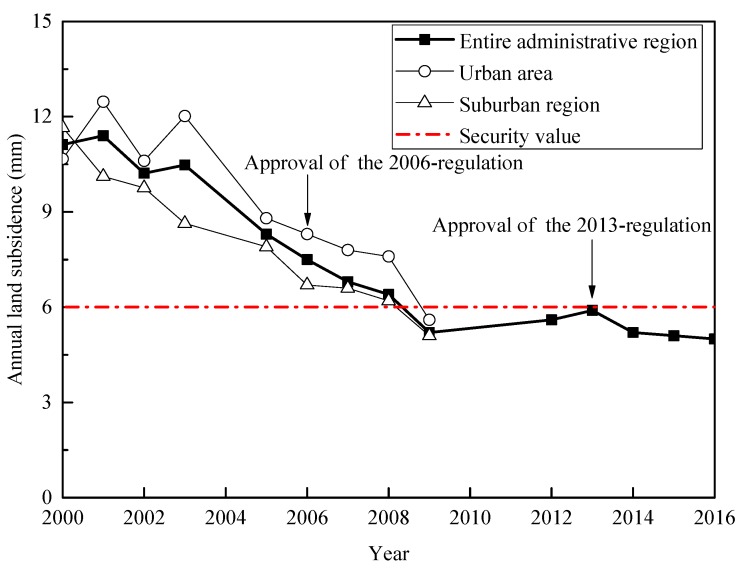
Monitoring results of annual land subsidence between 2000 and 2016 in Shanghai (data from Reference [26]).

**Table 1 ijerph-16-02729-t001:** Main information of land subsidence control zones in Shanghai (data from Reference [39]).

Name of Control Zones		Area (km^2^)	Distribution Areas	Control Objectives by the End of 2020
Key land subsidence control zones (I)	Key control sub-zones I_1_	649	Urban center within outer ring line	(1) To control the annual average land subsidence (AALS) within 7 mm; (2) to reduce the impact of differential land subsidence
	Key control sub-zones I_2_	1307	Pudong New District and Dahongqiao District beyond the outer ring line	(1) To control AALS within 7 mm; (2) to steadily raise the groundwater level; (3) to alleviate the impact of differential land subsidence
	Key control sub-zones I_3_	476	(1) Within 500 m on both sides of the major infrastructure protection areas such as high-speed rail and rail transit; (2) within 500 m of the flood control wall and the inner side of the seawall in the sub-key and general land subsidence control zones (zones II and zones III)	(1) To control AALS within 5 mm; (2) to reduce differential land subsidence along major infrastructure (including high-speed rail, rail transit, flood control wall, and seawall)
Sub-key land subsidence control zones (II)		974	Baoshan District, Jiading District and Minhang District	(1) To control AALS within 6 mm; (2) to basically eliminate of the depression cone of groundwater level around major infrastructure
General land subsidence control zones (III)		3903	Fengxian District, Songjiang District, Jinshan District, Qingpu District and Chongming District	(1) To control AALS within 5 mm; (2) to actively eliminate the depression cone of groundwater level

**Table 2 ijerph-16-02729-t002:** Main control measures of each control zones in Shanghai (data from Reference [39]).

No.	Main Control Measures by the End of 2020	Control Zones
1	(1) Coordinated estimation of the allocation of groundwater resources in the city and the groundwater withdrawal and recharge plan in the zones should be adjusted dynamically; (2) the total amount of groundwater withdrawal in the AqII to AqV will be controlled at *Q*_1_ million m^3^ per year; (3) the total amount of groundwater artificial recharge is not less than *Q*_2_ million m^3^ per year	I_1_ (*Q*_1_ = 0.8; *Q*_2_ = 10) I_2_ (*Q*_1_ = 1; *Q*_2_ = 1.4) II (*Q*_1_ = 0.7; *Q*_2_ = 3.2) III (*Q*_1_ = 5.5; *Q*_2_ = 5.4)
2	Special recharge of groundwater in shallow confined aquifers with emphasis on the AqI is implemented, to ensure that the artificial recharge of groundwater in the AqI reaches about 0.4 m^3^ per year	I_1_, I_2_
3	(1) Improve key safety technologies for land subsidence along major infrastructure areas; (2) reduce the impact of uneven settlement on major infrastructure	I_1_, I_2_
4	Deepening the research on the mechanism and control countermeasures of engineering land subsidence caused by the precipitation of foundation pit	I_1_, I_2_, II
5	Improve the prevention and measure of land subsidence based on the management of groundwater withdrawal and recharge	I_1_, I_2_, II
6	Improve the backbone monitoring network of land subsidence along major infrastructure such as rail transit and high-speed rail	I_2_, I_3_, III
7	Efforts should be made to strengthen the investigation and mechanism study of land subsidence in coastal new land areas	I_2_
8	(1) Improve the monitoring system during the construction around major infrastructure; (2) the existing groundwater withdrawal wells shall be closed within 500 m on both sides of major infrastructure; (3) the groundwater withdrawal shall be prohibited, and drilling of new wells shall be stopped	I_3_
9	(1) Strengthen the layered settlement analysis along major infrastructure; (2) deepen the study of land subsidence mechanism along major infrastructure; (3) improve the safety early warning mechanism of land subsidence along rail transit	I_3_
10	(1) The land subsidence investigation and mechanism research in the new land area of Hengsha Dongtan should be carried out; (2) the land subsidence monitoring stations should be deployed and constructed in the monitoring blank areas of the new land area and regional node towns	III
11	(1) In view of the relatively low water level areas of the AqIV and AqV formed by groundwater withdrawal in Qingpu Baihe and Jinshan Fengjing areas and adjacent areas of Jiangsu and Zhejiang province, the joint management of regional groundwater withdrawal should be continuously strengthened; (2) the investigation of the present situation of groundwater withdrawal in provincial boundaries and the inspection of irregular groundwater withdrawal should be jointly carried out	III

**Table 3 ijerph-16-02729-t003:** Specific evaluation of the 2013-regulation by SEA six principles.

SEA Principle	Establishment of the 2013-Regulation	Score Out of 5
1	Yes: Sustainability principles are considered in municipal management and urbanization plan.	5
2	Not perfect: Most of the examination just before the projects.	3
3	Yes: Groundwater withdrawal and constructions are considered for the cumulative impacts.	4
4	Yes: Specific sustainable measurements are conducted in major projects.	4
5	Yes: Monitoring of environment during construction and in operation period.	5
6	Not perfect: Limited public involvement before implementation.	3
Total score		24

Note: If SEA principles are fully and positively followed, a full score of 5 is awarded for each principle; the total score following SEA principles is 30.

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
