# Peer review of "Land Subsidence Control Zone and Policy for the Environmental Protection of Shanghai"

_ijerph, 2019, doi:10.3390/ijerph16152729_

Round 1
Reviewer 1 Report
In the manuscript, the authors introduced the history of land subsidence in Shanghai and reported the management policy in Shanghai. The paper is well organized and presented. However, the novelty of this paper is not clear. The characteristics and the set of the regulations are normal and have been well applied in other land subsidence zones. The authors are suggested to introduce the principles which the division of the control zone based on and the land subsidence status over the city area. What happened after RLS being issued? Please provide data and evidence, rather than only give the conclusions. Additionally, the development of land subsidence in different regions are different, due to the diversity of geological conditions, so what are the geological conditions in Shanghai? Can RLS in Shanghai be well introduced to other cities (the authors should provide evidence)? Figure 1, Table 1-2 are too wordy, please make them concise.
Author Response
Point 1: The authors are suggested to introduce the principles which the division of the control zone based on and the land subsidence status over the city area.
Answer: Thanks for the reviewer’s valuable comment. The division of the control zone took into consideration several factors, such as the geological background, urban development planning, land subsidence susceptibility and the development status of land subsidence. These have been added from line 172 to line 174. The land subsidence status over the city area have been added in the Section 2 from line 66 to line 69. Figure 1 shows the overall land subsidence in Shanghai from 1921 to 2010 in line 75.
Point 2: What happened after RLS being issued? Please provide data and evidence, rather than only give the conclusions.
Answer: Thanks for the reviewer’s valuable comment. The data and evidence have been provided in Section 5 (Discussion). According to the SEA analysis results, the score of the 2013-regulation in SEA was 24, while the score of the 2006-regulation in SEA was 16. You can find them from line 190 to line 220.
Point 3: Additionally, the development of land subsidence in different regions are different, due to the diversity of geological conditions, so what are the geological conditions in Shanghai?
Answer: Thanks for the reviewer’s valuable comment. The geological conditions have been added in Section 2 from line 69 to line 74. And geology and hydrogeology profile of Shanghai can be seen in figure 2 in line 77.
Point 4: Can RLS in Shanghai be well introduced to other cities (the authors should provide evidence)?
Answer: Thanks for the reviewer’s valuable comment. Shanghai is a city with plentiful water resources, such that land subsidence can be limited by the control of groundwater withdrawal and recharge. The 2013-regulation is formulated according to the characteristics of Shanghai city. The experience 2013-regulation of in Shanghai can be used for reference while it may not be directly applied in other cities. For example, it is inadvisable to limit the land subsidence by controlling groundwater withdrawal in North China Plain due to the shortage of water resources. These sentences were added in the end of Discussion (Section 5) from line 215 to 220.
Point 5: Figure 1, Table 1-2 are too wordy, please make them concise.
Answer: Thanks for the reviewer’s valuable comment. We have simplified the figure 1 (now is figure 4) and table 1-2 as much as possible.
Reviewer 2 Report
To improve the manuscript, i made all my comments, ask questions and suggest correction in the document itself (see comments in the pdf)

Author Response
For the review’s comments in the pdf file, we have revised the section according to the pdf file. The revised sections were highlighted in the word file. Several questions and answers are listed at the follow:
Point 1: Line 37: has decided or decided if you keep had decided then we are expecting another decision more recently taking into account past decisions? By the way when it has been decided? Is it still the case today?
Answer: Thanks for the reviewer’s valuable comment. After checking the reference, we think using “decided” is right. And the policy was decided at the turn of the century (since 2001). The paper we cited was written in 2006, so that we can’t know whether it is still the case today. We have revised this section from line 38 to line 39.
Point 2: Line39:explain the abbreviation InSAR-SB for the reader?
Answer: Thanks for the reviewer’s valuable comment. InSAR is the abbreviation of Interferometric Synthetic Aperture Radar, and SB is the abbreviation of small baseline interferometry which is one of the time-series techniques. We have revised this section from line 40 to 41.
Point 3: Line43:based on GPS stations was installed and what is the purpose of the 8 years, at least we don't care about this if they are no results associated?
Answer: Thanks for the reviewer’s valuable comment. We agree with your opinion. 8 years was just written in the reference and it may not be applicable for this paper, hence we have deleted “over 8-year of” in line 45.
Point 4: Line49: rephrase that sentence, unclear due to contradictory information.
Answer: Thanks for the reviewer’s valuable comment. We have rephrased the sentence from line 50 to 52.
Point 5: Line53:RLS means what? unclear here it's an abbreviation so R? is what i guess LS corresponds to Land Subsidence
Answer: Thanks for the reviewer’s valuable comment. RLS is the abbreviation of “the Regulations of the Shanghai Municipality on the Administration of Prevention and Control of Land Subsidence”. And R is the abbreviation of Regulations. Yes, it will be unclear to the readers. So now we don’t use the administration RLS, but use “the 2013-regulation” instead in the whole text.
Point 6: Line64:check everywhere in the text. There is a space between the numerical value and unit symbol, even when the value is used in an adjectival sense
Answer: Thanks for the reviewer’s valuable comment. We have checked the mistakes in the whole text.
Point 7: Line87:In figure 1, delete the dot of “No.62”. never mentioned in the text RGS? difference with RLS?
Answer: I'm sorry, it's a spelling mistake. We have deleted the dot of “No.62”. RGS is a spelling mistake. We have revised the mistakes in figure 1 (now is figure 4) in line 120.
Point 8: Line96:what do you mean by "complete" unclear? This monitoring network is based on what type of measurements, instrumentations, automatic, manual? on a monthly base, yearly base? how is working be more specific?
Answer: Thanks for the reviewer’s valuable comment. We think “improve” is more suitable than “complete”. And the monitoring network uses the technologies including GPS and GIS for automatically measuring land subsidence each year. We have revised this section from line 131 to 133.
Point 9: Line102:Administrative region limits, for both legends.
Answer: Thanks for the reviewer’s valuable comment. We have revised the legend in figure 5 in line 138.
Point 10: Line107:when, since it is in application?
Answer: Thanks for the reviewer’s valuable comment. The groundwater withdrawal is forbidden in the region covered by the pipeline network of water supply since the application of the 2013-regulation. We have revised this section from line 145 to line 146.
Point 11: Line111: “should” meaning that it is already in use or it's just a future plan?
Answer: Thanks for the reviewer’s valuable comment. The plan was carried out since the application of the 2013-regulation, so that it is already in use. We just tell the contents of the 2013-regulation, so we used “should”.
Point 12: Line115: why using plural? the only recognized geohazard is land subsidence?
Answer: Thanks for the reviewer’s valuable comment. The “geo-hazards” here contains a series of geo-hazards caused by land subsidence, such as loss of altitude, invasion of seawater, and urban waterlogging. We have revised this section from line 153 to 154.
Point 13: Line128: unclear? what do you means by fine standards?
Answer: Thanks for the reviewer’s valuable comment. “Fine standards” means the standards are detailed and complete. We think “perfect” is more suitable than “fine”. We have revised this section in line 168.
Point 14: Line139: Table 1, sentence unclear "the levels of the groundwater funnels?
same remark below as well
Answer: Thanks for the reviewer’s valuable comment. We have rephrased the unclear sentence in table 1 in line 180.
Point 15: Line141: Figure 3, two polylines are not represented in the legend? meaning? add them
Answer: Thanks for the reviewer’s valuable comment. The two polylines have been represented in the legend in figure 3 (now is figure 6) in line 182.
Point 16: Line147: Table 2 line 1, this is the first time that the different aquifers are mentioned in the document, it would have been good to add a small chapter to add this geological information and to add a reference for the reader
Answer: Thanks for the reviewer’s valuable comment. The geological and hydrogeological conditions have been added in the Section 2 from line 69 to line 74. And geology and hydrogeology profile of Shanghai can be seen in figure 2 in line 77.
Point 17: Line147: Table 2 line 5, unclear sentence, rephrase
Answer: Thanks for the reviewer’s valuable comment. We have rephrased the unclear sentence in line 5 of table 2.
Point 18: Line147: Table 2 line 8, strange sentence, rephrase.
Answer: Thanks for the reviewer’s valuable comment. We have rephrased the strange sentence in line 8 of table 2.
Point 19: Line147: Table 2 line 9, where are these districts? not represented on map
Answer: Thanks for the reviewer’s valuable comment. Three areas including Hengsha Dongtan、Qingpu Baihe and Jinshan Fengjing have been represented in figure 6 in line 182.
Point 20: Line147: Table 2 line 11, outside the map area and Shanghai districts, where are these provinces?
Answer: Thanks for the reviewer’s valuable comment. The Zhejiang and Jiangsu Province have been added on the map in figure 6 in line 182. And the East China Sea Yangtze River and Hangzhou Bay have also been added on the map in figure 6 in line 182.
Point 21: Line150: why eventual? rephrase the sentence unclear
Answer: Thanks for the reviewer’s valuable comment. It means we use SEA to evaluate the 2006-regulation, but the score is not very high. Hence, we have now used “eventual result” instead of “eventual score”. We have revised this section in line 193.
Point 22: Line153: PLS? what represents this abbreviation, never mentioned before in the text?
Answer: Thanks for the reviewer’s valuable comment. Now we don’t use the abbreviation PLS, but use “the 2006-regulation” instead. We have revised this section in line 193 and 195.
Point 23: Line169: in mm between 2000 and 2016.Why limited to 2016, no values recorded for the last 3 years?
Answer: Thanks for the reviewer’s valuable comment. The data is from Shanghai institute of geological survey which is just update to 2016. Therefore, we can’t find the new data of 2017 and 2018.
Reviewer 3 Report
see attached file

Author Response
Thanks for the reviewer’s valuable comment. Figure 1 was added to show the map of LS along the coast of Shanghai. Discussion are conducted to show the interaction between LS and sea level rise (Line 65 to 90). The suggested publications are cited.
Point 1: The paper is interesting but it lacks of maps with the distribution and rates of LS. In particular are not discussed the effects of LS along the coast of Shanghai.
Answer: Thanks for the reviewer’s valuable comment. The distribution of LS in Shanghai have been shown in figure 1 in line 75. The effect of LS along the coast of Shanghai have been added from line 80 to line 90. And the control measures of LS along the coast of Shanghai have already been discussed in section 4.4. The areas along the coast of Shanghai belong to key control sub-zones I3 and corresponding control measures are listed in table 2 in line 189.
Point 2: Which are the subsiding coastal zones most prone to marine flooding? In addition, can the authors provide sea level rise scenarios based on the ongoing climate change projections released by the IPCC (www.ipcc.ch and references below)? To this goal are required additional analyses and modeling for environmental interpretations and improve the scientific content of the paper.
Answer: Thanks for the reviewer’s valuable comment. Based on the ongoing climate change, the global average sea level from 1990 to 2009 have been added in figure 3 in line 91. And the sea level status along Shanghai have discussed from line 84 to line 86. Our paper is just a case report which introduces the land subsidence policy and control measures in Shanghai. Therefore, additional analyses and modeling for environmental interpretations will be a topic for future research.
Round 2
Reviewer 1 Report
The raised comments have been addressed. However, the novelty and the significance of this paper should be better demonstrated.
Author Response
The raised comments have been addressed. However, the novelty and the significance of this paper should be better demonstrated.
Answer: Thanks for the reviewer’s valuable comment. In 1996, a regulation for the management of land subsidence monitoring network (labeled as 1996-regulation) was approved by the municipal government of Shanghai. In 2003, the urban planning (labeled as 2003-planning) proposed by Shanghai Municipal Planning and Ground & Resources Administration, in which land subsidence control aim was proposed. In 2006, the Regulation of Prevention and Control of Land Subsidence of Shanghai Municipality (labeled as 2006-regulation) was implemented. However, these regulations and planning (1996-regulation, 2003-planning, and 2006-regulaton) did not completely follow the principles of the strategic environmental assessment (SEA) for sustainable urban development [23]. Most of the principles of SEA were just partially considered in the 2006-regulation. Considering the deficiency of 2006-regulation, the Regulation of Prevention and Control of Land Subsidence of Shanghai Municipality in 2013 (labeled as 2013-regulation) was issued [24, 25]. To ensure the sustainable development of Shanghai, implementation effect of the 2013-regulation should be evaluated after 6 years application of the 2013-regulation. The SEA results shows that land subsidence value was controlled within a small level. However, more effort should be conducted in the implementation of SEA in land subsidence control in future. Thus, the novelty and the significance of this paper is to reveal the effectiveness of SEA in municipal’s environmental management. These have been added from line 27 to 29 in abstract, from line 57 to line 70 in introduction, from line 201 to line 202 in discussion and from line 238 to line 239 in conclusions.
Furthermore, we have accepted the editors’ revision in the manuscript. Thanks to the editors for the correcting the article.